# Arms Race between the Host and Pathogen Associated with Fusarium Head Blight of Wheat

**DOI:** 10.3390/cells11152275

**Published:** 2022-07-23

**Authors:** Chunhong Hu, Peng Chen, Xinhui Zhou, Yangchen Li, Keshi Ma, Shumei Li, Huaipan Liu, Lili Li

**Affiliations:** 1College of Life Science and Agronomy, Zhoukou Normal University, Zhoukou 466000, China; ourcarrot@163.com (C.H.); pengchen2732@163.com (P.C.); zxh2452826493@163.com (X.Z.); yc1524373047@163.com (Y.L.); zknumks@126.com (K.M.); henanlsm@126.com (S.L.); 2Key Laboratory of Plant Genetics and Molecular Breeding, Zhoukou Normal University, Zhoukou 466000, China

**Keywords:** wheat, fusarium head blight (FHB), resistant germplasm resources, pathogenesis, resistance mechanism, resistant QTL/genes, signaling pathway

## Abstract

Fusarium head blight (FHB), or scab, caused by *Fusarium* species, is an extremely destructive fungal disease in wheat worldwide. In recent decades, researchers have made unremitting efforts in genetic breeding and control technology related to FHB and have made great progress, especially in the exploration of germplasm resources resistant to FHB; identification and pathogenesis of pathogenic strains; discovery and identification of disease-resistant genes; biochemical control, and so on. However, FHB burst have not been effectively controlled and thereby pose increasingly severe threats to wheat productivity. This review focuses on recent advances in pathogenesis, resistance quantitative trait loci (QTLs)/genes, resistance mechanism, and signaling pathways. We identify two primary pathogenetic patterns of *Fusarium* species and three significant signaling pathways mediated by UGT, WRKY, and SnRK1, respectively; many publicly approved superstar QTLs and genes are fully summarized to illustrate the pathogenetic patterns of *Fusarium* species, signaling behavior of the major genes, and their sophisticated and dexterous crosstalk. Besides the research status of FHB resistance, breeding bottlenecks in resistant germplasm resources are also analyzed deeply. Finally, this review proposes that the maintenance of intracellular ROS (reactive oxygen species) homeostasis, regulated by several TaCERK-mediated theoretical patterns, may play an important role in plant response to FHB and puts forward some suggestions on resistant QTL/gene mining and molecular breeding in order to provide a valuable reference to contain FHB outbreaks in agricultural production and promote the sustainable development of green agriculture.

## 1. Introduction

Fusarium head blight (FHB) is one of the most devastating and difficult fungal diseases in the world. It is caused by *Fusarium* species and is known as the “cancer” of wheat. FHB not only severely reduces wheat yield but also leads to the accumulation of various toxins, such as deoxynivalenol (DON), nivalenol (NIV), and zearalenol (ZEN), in wheat kernels infested by *Fusarium* species [1,2]. Therefore, once these toxins enter the bodies of humans and animals, they cause serious harm to human and animal health. Wheat is one of the most widely planted crops in the world, and the sustainability and stability of wheat production are directly related to the issue of food security. Thus, FHB has become one of the most severe diseases that need to be addressed urgently in wheat production.

In recent years, with the influence of global warming and changes in wheat farming systems and agricultural production techniques, the occurrence of FHB has become increasingly serious in wheat-producing regions around the world, such as Asia, North America, South America, and Europe. [3]. Therefore, FHB burst have seriously threatened the development of wheat production and have become the focus of scientists’ attention worldwide. Scientists have made extensive efforts in many aspects, such as selection and identification of FHB-resistant germplasm resources [4,5]; genetic breeding [6,7,8,9,10]; biological characteristics and pathogenesis of *Fusarium* species [11,12,13,14,15]; epidemic occurrence pattern and mechanism [16,17]; disease identification [18]; and integrated prevention and control technology [19,20,21]. Especially in the era of molecular biology, scientists have made significant progress in the localization, cloning, and function of FHB-resistance genes and molecular breeding to resist FHB [22,23,24,25]. However, most of the resistance resources identified have not been well researched and utilized so far. Although some FHB-resistance loci or genes have been discovered, their functions just have been characterized, and their stability and validationare rarely reported, which seriously affect the process of breeding. For these reasons, the application of fungicides is currently the most popular method to control FHB, but this will inevitably increase the cost of farming and environmental pollution. At the same time, biological control methods, due to the advantages of no environmental pollution, have received more and more attention. However, the volatile living environments and the variability of inherited traits of microorganisms become the limiting factors for their popularization and application in production. Thus, cloning and dissecting the functions of resistance genes related to FHB, and breeding FHB-resistant germplasm resources, will be the most effective ways to control FHB outbreaks and promote sustainable agricultural development.

## 2. Identification of Pathogenic Strains of Wheat FHB and Its Pathogenesis

To date, more than 20 strains of *Fusarium* or their variants have been identified worldwide. With the rapid development of molecular biology and detection technology, experts in FHB research isolated the pathogenic strains in nature by means of QTL mapping and gene localization techniques [6,14,26]. The pathogenic strains and their pathogenicity were identified and assessed by manual single-flower drip inoculation and metabolic toxin detection. It was found that, in China, *Fusarium graminearum*, *Fusarium culmorum*, and *Fusarium avenaceum* were all of the pathogenic strains with high and lethal pathogenicity [27] (pp. 3–12). Among these *Fusarium* species, *F. graminearum* (*Fg*) is considered the strongest pathogenic strain, due to its strong viability and wide range of hosts. Scholars have carried out much research on the infestation process and the pathogenesis of *Fusarium* species and have found that *Fg* infects not only the above-ground parts but also the below-ground parts of the plant [15]. Moreover, it generally invades plants in two ways. One is direct stretching; the mycelium penetrates into the host organism through natural pores, such as stomata, apical pores of florets, and slits at the junction of the palea and glumes [28]. The other is direct penetration; the mycelium enters into plants through the epidermal cuticle, middle lamellae, and cell walls (Figure 1) [29]. For example, Xu and Hideki (1989) found that *Fusarium* firstly invaded host anthers through the epidermal cells with mycelium, and then the spores germinated into mycelia which expanded laterally towards the glumes, including the inner and outer glumes. [30].

With the advance in research, emerging evidence has demonstrated that many pathogenic factors and metabolites from pathogenic fungi or plants are involved in the process of host–pathogen interactions. First of all, ROS bursts are a common feature of plant response to external stresses [31,32], and stronger ROS bursts in the later infection stage promote pathogen proliferation and toxin production [33]. Secondly, both choline analogs and neutral betaine (alkaline substances in wheat anthers) were reported to promote the growth and development of *Fg* [34,35]. Endo-polygalacturonase (endo-PG) [27] (pp. 3–12) and xylanolytic and glucanolytic enzymes [36] also play crucial roles in fungi invading and degrading the cell walls of the host. A transduction beta-like gene, *FTL1*, is essential for pathogenesis on wheat spikes [37]. A linear octapeptide from *Fg* can facilitate the spread of mycelia through the cell wall to neighboring cells [34]. In addition, GTP binding protein and G protein-coupled receptor complexes are also essential for signal transmission during the early infection process of *Fg*. For example, the heterotrimeric G protein subunits Gα, Gβ, and Gγ [38], Ras-GTPase RAS2 [11], G protein-coupled receptor Ste2 [39], dynamin-like GTPase protein Sey1 [40], and ADP-ribosylation factor-like small GTPase Arl1 [41] are all involved in DON biosynthesis and pathogenesis. Furthermore, the mitogen-activated protein kinase (MAPK) STE7/11 and GPMK1depended cascades, being downstream of Ras-GTPase RAS2, are all involved in the secretion of xylanases, proteases, endoglucanases, and lipases in *Fg* [36]; a Rab5 guanine nucleotide exchange factor (GEF), Vps9, from *Fg*, by interacting with the guanosine diphosphate (GDP)-bound (inactive) forms of Rab51 and Rab52, plays an instrumental role in vegetative growth, asexual development, autophagy, DON production, and plant infection in *Fg* [42]. Moreover, an ATPase component, Rad50, also plays a crucial role in fungal development, virulence, and secondary metabolism in *Fg* as well as cell wall integrity and the DNA damage response [43]. Serine/arginine (SR) proteins Srp1 and Srp2 synergistically control the vegetative growth, sexual reproduction, and pre-mRNA processing in *Fg* [44]. Moreover, a pyruvate dehydrogenase kinase—PDK2 [45], a SNARE protein FgSec22 [46], a purine nucleotide IMP/AMP/GMP [47], and a phosphatidylserine decarboxylase FgPsd2 [14] are all instrumental in vegetative growth, pathogenesis, and DON biosynthesis in *Fg*.

From the results mentioned above, it could be easily concluded that FHB pathogenic fungi should first enter the plant through natural pores or a wound and then synthesize some metabolites to facilitate the growth and development of mycelium. In order to further expand in hosts from one cell to another, pathogenic fungi synthesize some pathogenic factors and/or some hydrolytic enzymes to degrade the host cell wall and break through the plant’s intracellular physical defense system by initiating signaling pathways. Among these pathways, the small GTPase (Rab/Ras)-GEF-MAPK-mediated signaling process is relatively clear, even though its molecular mechanism remains to be further elucidated. The arms race is always fierce between the pathogen and its host. The host always performs a resistance defense system to inhibit the invasion and expansion of pathogenic fungi. For instance, in the early stage of pathogen infection, plants synthesize chitinases, tylopectinase, and glucanase to degrade the fungal cell wall and produce a rapid ROS burst and synthesize thionins, non-specific lipid transfer proteins, and puroindolines to disrupt fungal membrane integrity (Figure 1❶,❷). With ROS accumulating in plant cells, plants synthesize oxidases, such as peroxidase (APX), superoxide dismutase (SOD), and catalase (CAT), to scavenge ROS, thus reducing oxidative damage for themselves (Figure 1❷,❸,❸). However, in the later stage of infection, stronger ROS bursts are induced by pathogenic factors, which, in turn, promote disease spread and toxin accumulation and then result in programmed cell death in plants [33] (Figure 1❹).

## 3. Discovery and Identification of Wheat Resistance Germplasm Resources

Germplasm resources are the basis of disease-resistance breeding. Since the 1920s, countries such as China, the USA, Korea, Japan, Argentina, Brazil, Switzerland, and the Czech Republic have made unremitting efforts in the identification and discovery of germplasm resources for resistance to FHB [3]. Since the outbreak of FHB in China in 1936, Chinese scientists have made great efforts in screening resistant germplasm resources. In 1974, the China Corporation of Research on Wheat Scab (CCRWS) carried out a nationwide screening of germplasm resources for resistance to FHB and identified 34,571 materials, including the common and wild wheat relatives. Only 1796 common wheat varieties were recognized as high- or medium-resistant resources to FHB, including the accepted “Wang Shui Bai” and “Su Mai 3” [5]. In 1997, Wan et al. identified 276 materials of 80 species in 16 genera of wheat relatives and found that *Roegneria* (*Roegneria tsukushiensis* var. *transiens* and *ciliaris*) was the most resistant; *Elymus*, *Kengyilia*, *Agropyron*, *Elytrigia*, and so on were moderately resistant; *Aegilops*, *Crithopsis*, and *Eremopyrum* were susceptible [48]. However, Gagkaeva (2003) analysed nine species of *Aegilops* from warm and humid areas and found that *Aegilops* was a potential source of FHB resistance [49]. In addition, Brisco et al. (2017) identified more than 99 *Aegilops* species from areas with high levels of annual rainfall and further verified that *Aegilops tauschii* Coss was FHB-resistant [4]. Obviously, the differences in the results mentioned above may be attributed to the different research conditions. Gagkaeva (2003) identified the FHB resistance of 252 materials and found no relationship between ploidy and FHB resistance but a close correlationship between FHB resistance and the geographical origin of the materials. Wheat relatives from high-warmth and -humidity environments were FHB-resistant, while the resources from dry environments in Central Asia were highly susceptible to FHB [49]. With further research, more and more FHB-resistant resources have been found or identified again, such as *Leymus racemosus* Tzvelev (*Elymus giganteus* Vahl.) [50], *Elymus tsukushiensis* honda [51], *Elytrigia elongata* (Host) Nevsi [52], and so on. The results mentioned above demonstrate that FHB resistance of wheat ishows specificity in species and environments. The wheat wild relatives, such as *Roegneria*, *Leymus racemosus*, *Elymus tsukushiensis*, *Elytrigia elongata*, and *Aegilops*, are all potential FHB-resistant germplasm resources. Moreover, FHB resistance is closely related to the geographical origin of the materials in warm and humid areas.

In summary, more than 50,000 wheat materials have been identified worldwide for resistance to FHB in the last 10 years. According to the incidence of FHB on wheat spikes, the severity of the disease was divided into five grades: 0, I~IV—from disease-free to mild to severe. Consistent with this, wheat germplasm resources are also classified into five grades according to their resistance to FHB (Figure 2). However, it is clear that, so far, almost all of the FHB-resistant germplasm resources selected globally are moderately susceptible, and none of them is completely immune to FHB [3]. The better resources, which had been identified as having good resistance to infestation and extension, were mainly found in the middle and lower reaches of the Yangtze River in China, a region with a high incidence of FHB. For example, the resistant materials Yang Mai 158, Su Mai 3, Wang Shui Bai, Ning Mai 9, and dozens of wheat varieties derived from them all originated in these regions [53,54,55], while only Sumai 3 and its derivatives have been widely used for FHB-resistance breeding in China and abroad and have resulted in resistant varieties such as Ning 7840 [6], Alsen [7], CM 82,036 [9], and Saikai 165 [8]; most other highly resistant FHB germplasm resources have not been promoted for production and breeding, due to their poor agronomic traits. Even though several varieties have been promoted for production, only very few reach moderate resistance levels (e.g., Yang Mai 158 and Ning Mai 9), and very few reach high resistance levels [55]. In conclusion, the identification of FHB-resistant resources has made an important contribution to FHB resistance research worldwide, and a number of resistant germplasm resources have been discovered. However, the resistance levels of the existing resistant varieties still cannot compensate for the large yield losses in years of severe blast epidemics. Therefore, it is imperative to explore high-resistance germplasm resources and, especially, to excavate resistance materials from high-humidity and -rainfall environments.

## 4. The Resistance Mechanism of Wheat to FHB

### 4.1. Morphological and Physiological Mechanisms of Wheat Response to FHB

Where there is aggression, there is resistance. Study on the resistance mechanism of wheat to *Fg* is of great importance in controlling FHB and has become a major focus of attention for researchers. Based on the phenotype of resistance to FHB, plants have also been classified into five types: Type I, resistance to the initial infection; Type II, resistance to proliferation; Type III, seed resistance to infection; Type IV, tolerance to disease; and Type V, resistance to toxin accumulation [56,57]. All these resistance types can interact with each other to synergistically improve the overall resistance of wheat [58]. Among them, Type I and Type II have been more intensively studied, mainly in terms of the morphological and physiological mechanisms [59]. For example, many studies have demonstrated that the morphological characteristics—plant height, length of spike and flowering period, degree of anther extrusion, presence of awn, spike length and density, degree of glume opening, and degree of waxiness of the spike—are all correlated with FHB resistance against invasion [60,61]. More importantly, cytological studies showed that the pathogen-resistant varieties synergistically inhibited the expansion of the pathogen through forming papillae, reinforcing cell wall deposits, and increasing the biosynthesis of lignin, thionine, hydroxyproline-rich glycoprotein, and hydrolase [62]. In addition, Liu et al. (2022) suggested that plants dynamically regulate stomatal reopening, mediated by the secreted peptides SCREWs and the receptor kinase NUT, to ensure a balanced physiological response at the whole-plant level in response to biotic stress [63].

It is well known that, when invaded by pathogenic fungi, plants synthesize different signaling molecules that stimulate the expression of disease-resistant genes through complex signaling pathways and thereby resist the invasion of pathogenic fungi. With progressive research, three hormone-mediated signaling pathways, jasmonic acid (JA), salicylic acid (SA), and ethylene (ET), have come to be viewed as widely involved in plant biotic stresses [36,64,65]. The positive role of JA in FHB resistance has been demonstrated by studies [66,67]. For instance, a pore-forming toxin-like protein, PFT, may play a role in JA-mediated FHB resistance in a resistant wheat cultivar [68,69]. A wheat allene oxide synthase, TaAOS, is also involved in the JA signaling pathway to increase plant resistance to FHB, and the silenced strains exhibit a highly susceptible trait [70]. However, the roles of SA and ET in FHB resistance remain to be further demonstrated [71,72]. Furthermore, the specific signaling pathways involved in the response of the three hormones to FHB stress in plants have not been clarified. Nevertheless, some progress has been made, especially in the basic physical and physiological defense carried out by plants in response to FHB stress (Figure 1). However, FHB is caused by a mixture of *Fusarium* species and affected by genetic and environmental factors. FHB resistance is a quantitative trait that is controlled by multiple quantitative trait loci (QTL). Therefore, the number of resistance master genes which have been identified is still very limited, and the resistance mechanism is still unclear [73], which has seriously affected the research process of FHB resistance improvement.

### 4.2. Resistance QTL in Plant to FHB

Analyzing the genetic loci of resistant germplasm resources and learning about their hereditary features is a prerequisite for applying high-quality resources to wheat genetic breeding. Numerous studies have shown that FHB resistance is a quantitative trait and is controlled by multiple genes. Early genetic studies were carried out by segregating progeny, estimating the resistance genes that different resistance parents might carry and the chromosomal localization of genes through chromosome engineering techniques. To date, many of the FHB-resistant QTLs in the representative resistance germplasm resources have been mapped onto all 21 wheat chromosomes (Table 1) [74]. Several important resistant seed resources of FHB have been used for chromosome mapping and functional analysis of QTLs by creating genetic populations such as the recombinant inbred lines (RIL) or double haploid lines (DHL). The advent of DNA markers and marker-based genetic mapping in the 1990s greatly facilitated the fine targeting of quantitative trait loci (QTL) or genes on genetic maps. For example, Li et al. (2019) finely localized the genetic locus of *Qfhb.nau-2B* in Nanda 2419 [75], and Jiang et al. (2020) localized the FHB resistance QTL *QFhb-5A* in Yangmai 158 [55]. In addition, the fungus-resistant locus in different wheat resistance strains, such as the Swiss wheat variety Arina [76]; Ernie [77] and Truman [78] in the USA; Dream [79] in Germany; T. macha [80] in Maga, Frontana [81] in Brazil; Chok-wang [82] in Korea; and Nyubai [83] and SYN1, a synthetic species bred by the International Maize and Wheat Improvement Center (CIMMYT), all have been identified by constructing genetic populations RILs or DHLs. Furthermore, with the application of bioinformatics technology and the continuous improvement of genomic data, the methods of genome-wide association studies (GWAS) have been used to excavate resistant loci/QTLs or associated genes [84]. To date, hundreds of FHB resistance QTLs have been identified on the 21 wheat chromosomes [74,85,86]. However, most of them have minor or unstable effects, or their genetic loci and the molecular markers vary from one experiment to another. Moreover, the QTLs associated with type I resistance have low reproducibility due to different infection and identification methods and environmental conditions. Therefore, only a handful of QTLs, such as *Fhb1*, *Fhb2*, *Fhb4*, *Fhb5*, and *Fhb7*, have been finely localized and successfully employed in breeding programs [87], and very few QTLs associated with extension resistance have been finely localized and used in breeding. For detailed information on the superstar QTLs related to FHB resistance, please refer to Table 1.

Based on traditional breeding, molecular marker-assisted selection (MAS) is regarded as a useful tool for breeding, and it indeed improves the efficiency of selective breeding. However, due to linkage drag, the introduction of the FHB resistance QTLs is often accompanied by undesirable traits, which leads to certain difficulties for later genetic breeding.

### 4.3. Resistance Genes and Their Mechanisms in Response to FHB in Plant

So far, although hundreds of QTLs for FHB resistance have been mapped onto wheat chromosomes, it is a challenging task to segregate them using forward genetics. At present, the more well-defined QTLs for resistance to FHB in wheat are *Qfhb.nau-2B* [75], *Fhb1~Fhb7* [85], *Qfhs.ndsu-3AS* [88], *QFhb-5A* [55], and so on (Table 1). Among them, the first identified and the best validated resistance QTL is *Fhb1,* which is employed as the most important FHB resistance donor worldwide [52,89]. Therefore, several approaches have been carried out to identify candidate genes in *Fhb1* by transcriptome-based analysis and map-based cloning. For example, the genes encoding a pore-forming toxin-like proteinPFT [68,69], a laccase TaLAC4 [100], and a NAC (N-acetylcysteine) class transcription factor TaNAC032 [101] were all positioned within *QTL-Fhb1* and conferred to FHB resistance. In addition, two putative histidine-rich calcium-binding proteins, TaHRC and *Qfhs.njau-3B*, were also found underlying in *QTL-Fhb1*, but they conferred FHB susceptibility. On the contrary, mutants of them conferred FHB resistance [52,89].

The advancement of genomics, proteomics, and metabonomics technologies has provided opportunities for identifying candidate genes and resolving the complex genetic mechanisms of FHB resistance. Therefore, a number of FHB-resistance genes have been reported (Figure 3 and Table 2). First of all, several uridine diphosphate (UDP)-glucose transferases (UGTs) in wheat had been reported to be involved in FHB resistance in plants via cellular detoxification processes [98,102,103]. For example, TaUGT3 positively regulates the defense responses to FHB by converting DON to the less toxic DON-3-O-glucoside [104]; TaUGT4 was strongly induced by *Fg* or DON treatment according to transcriptional analysis [73]; Zhao et al. (2018) also proposed that TaUGT5 could reduce the proliferation and destruction of *Fg* and enhance the ability of FHB resistance in wheat [103]; another wheat TaUGT (Traes_2BS_14CA35D5D) was confirmed to enhance plant resistance to FHB and tolerance to DON as well as to potentially conjugate DON into D3G [105]. Later, a novel UGT gene, TaUGT6, was cloned and identified to enhance plant resistance to *Fusarium* by converting DON into D3G to some extent in vitro [106]. Secondly, another detoxification protein aglutathione S-transferase (GST), Fhb7, from a wheat relative, *Elytrigia elongata*, was reported to confer broad resistance and tolerance to *Fusarium* species by detoxifying trichothecenes, such as NIV, other than DON [95]. These results showed that detoxification mediated by uridine diphosphate (UDP)-glucose transferases (UGTs) and glutathione S-transferases (GSTs) is a crucial response of plants to resist FHB stress (Figure 3❶).

Additionaly, there are some other FHB-resistance genes that have been identified. For instance, a transcription factor, TaWRKY45, was proved to enhance resistance to FHB in wheat, though the mechanism of resistance is still unknown [119,120]. TaWRKY70 was identified within the FHB-resistant *QTL-2DL* region and had a potential role against *Fg* in the early stages of defense through physically interacting with and activating the downstream genes *TaACT*, *TaDGK*, and *TaGLI*, among which *TaACT* was the rate-limiting enzyme in the biosynthesis of HCAAs (hydroxycinnamic acid amides), while *TaDGK* and *TaGLI* were the crucial enzymes in the biosynthesis of PAs (phosphatidic acids) in plants [108]. It should be added here that both HCAAs and PAs are instrumental in fungus–plant interactions. To be specific, PAs contribute to the plant defense response through translocation and catabolism in the apoplast, leading to the production of H_2_O_2_, which has several roles, such as directly eliminating pathogens and assisting in cell wall strengthening; HCAAs were identified as biomarkers during fungus–plant interactions and support a functional role in plant defense by reinforcing cell walls [109]. What is more, a guanidine cinnamyl transferase gene, TaACT, was also confirmed to have a role involved in FHB response in plants [110]. Similarly, in barley, HvWRKY23, induced by a chitin elicitor receptor kinase HvCERK1 [123], also fights FHB by regulating its downstream genes, such as *HvPAL2*, *HvCHS1*, *HvHCT*, HvLAC15, and *HvUDPGT*, to biosynthesize HCAAs and flavonoid glycosides [124]. Moreover, a potential downstream gene, *TaLAC4*, encoding a wheat laccase, was reported to restrain FHB infestation by promoting the synthesis of lignin in the secondary cell wall of plants in order to thicken the cell wall [100]. In addition, it is well known that the first layer of innate immunity in plants is initiated by the surface-localized pattern recognition receptors (PRRs), in which the leading role is played by chitin elicitor receptor kinase 1 (CERK1), which mainly participates in chitin-induced immunity [107]. Meanwhile, chitin is a typical component of the fungal cell wall and can trigger the plant innate immune response by activating PAMPs (pathogen-associated molecular patterns). Therefore, a signaling pathway of CERK1-WRKY-mediated RRI metabolite accumulation in cells perhaps plays an important role in plant response to FHB stress (Figure 3❷).

Approaches are different, but the results are satisfactory. A NAC-like transcription factor, TaNAC032, was also reported to enhance FHB resistance by reinforcing the secondary cell wall of plants through regulating resistance-related proteins and metabolites such as phenylpropanoid and lignin [101]. Meanwhile, another NAC (N-acetylcysteine), which was classed as a transcription factor, TaNACL-D1, can positively regulate the expression of a wheat orphan protein, TaFROG, and improve plant resistance to FHB [113]. On the other hand, TaFROG can interact with the FHB-resistance protein TaSnRK1α to protect it from degradation by deubiquitination, which, in turn, confers plant resistance to the fungal mycotoxin DON [125,126]. Interestingly, Jiang et al. (2020) found that an orphan protein, Osp24, which is secreted from FHB-causing fungi, competes against TaFROG for binding with the same region of TaSnRK1α and leads to the degradation of TaSnRK1α by ubiquitination [117]. To date, many studies have demonstrated that SnRK1, as the metabolic/energy sensor and signaling integrator, is involved in the plant response to diverse stress and energy conditions [111,127], though the resistance mechanism of TaSnRK1α to FHB in plants still remains unclear. Thus, the previously mentioned information indicates that NAC-TaFROG/Osp24-TaSnRKα-mediated RRI metabolites perhaps also perform crucial roles in the plant response to FHB stress through regulating energy-related metabolites, such as phenylpropanoid and lignin, which are also utilized in strengthening cell walls (Figure 3❸).

Besides the aforementioned genes/proteins, lipid transferase protein LTP [98] and its interacting protein TaRBL [115], ARF-GEF protein MIN7 [112], allene oxide synthase gene *TaAOSb* [70], histidine-rich calcium-binding protein TaHRC [52], putative membrane protein WFhb1-1 [122], wheat-wall associated kinases TaWAK2A-800 [121], serine hydroxymethyltransferase TaSHMT3A-1 [116], stearoyl-acyl carrier protein fatty acid desaturase TaSSI2 [118], Jacalin-related lectins protein TaJRL53 [114] were all excavated and proved to be positively involved in FHB response in plants (Table 2). In summary, the results mentioned above urged us to determine the pathways by which the FHB-resistant genes were involved in fungus–plant interactions. From Figure 3, we can see that there are three resistant pathways: TaUGT/Fhb-GST-mediated detoxicated pathway and TaWRKY- and TaSnRK1α-mediated metabolic pathways, respectively.

To sum up, with the advancement of biotechnology, more and more QTLs/genes for FHB resistance loci have been identified or clearly localized on the chromosomes. However, to date, only a few genes have been studied in depth; most genes were just functionally characterized, and very few resistance mechanisms and signaling pathways were pinpointed and explained. In addition, FHB is a complex quantitative trait, and a single gene phenotypic contribution ranges from 15% to 30% [54]. Thus, relying on only one, two, or very few resistance genes is not sufficient to effectively reduce damage, especially in years of severe epidemics. Therefore, an effective way, for restraining FHB burst, is to breed the resistant germplasm resources by unearthing multiple and pivotal FHB-resistance genes, revealing their resistance mechanisms, and then aggregating them and their synergistic resistance-related genes into varieties (lines) with better fecundity.

### 4.4. CERK1-Mediated ROS Homeostasis Perhaps Performs a Vital Role in Response to FHB in Plant

It is well known that ROS bursts are a common feature of plant response to external stresses [31,32], especially in the earlier stage of pathogen infection [11]. NADPH oxidase (NOX), also known as respiratory burst oxidase homolog (RBOH), which is a key enzyme for ROS (·O_2_^−^, ·OH, H_2_O_2_) production in the intercellular matrix, has become the focus of current research on plant response to external stress. Previous studies on model plants have revealed that NOX is involved in a variety of signaling pathways which endow NOXs/RBOHs with powerful and versatile functions in plants to maintain innate immune homeostasis [128]. Moreover, many studies have indicated that some chitin-induced receptor-like kinases (RLKs) and/or receptor-like cytoplasmic kinases (RLCKs) can interact with NOXs/RBOHs directly or indirectly and phosphorylate the proteins to transmit pathogen signals during plant immunity [129,130]. Furthermore, several RLKs (CERK, LysM, and OsCEBiP) and RLCKs (BIK1 and small GTPase ROPs/RACs) and GEF-mediated immune signalings are all involved in ROS homeostasis by directly activating NOX in plants [107]. As early as 2011, Ding et al. found that the expression activity of NOX was significantly increased in the resistant variety “Wang Shui Bai” under FHB stress treatment but not in the susceptible varieties [131], suggesting that wheat NOX family members may play important roles in plant response to FHB stress. Accordingly, it is tempting to speculate that the chitin-induced particles CERK1, LYK, CEBiP, BIK1, and ROPs/RACs and GEF-mediated ROS homeostasis, by regulating NOX activity, perhaps perform a vital role in the plant response to FHB (as shown in Figure 4). In Arabidopsis, CERK1, a chitin elicitor receptor kinase 1, and LYK5, a LysM-containing receptor-like kinase 5, were found to be able to bind chitin and form a chitin-dependent complex, CERK1-LYK5 [132,133]. This is an immune complex in which AtCERK1 is involved in activating BIK1 by phosphorylation directly. After this, the activated BIK1 (GTP-BIK1) directly phosphorylates AtRbohD and enhances ROS production for defense responses [134,135]. In rice, OsCERK1, a plasma membrane-localized receptor-like kinase, is associated with receptor-like kinase OsCEBiP under chitin treatment [136]. In addition, OsCEBiP can form a homodimer upon chitin binding that is followed by heterodimerization with OsCERK1, creating a signaling-active sandwich-type receptor system [137,138]. Then, OsGEF1, a regulator functioning as a molecular switch, is phosphorylated by OsCERK1 [139]. The activated OsGEF1, coupling with OsRACK1 [140] in turn, phosphorylates a ROP/RAC GTPase, OsRAC1, and changes it from the RAC-GDP inactive form to the RAC-GTP active form [130,141]. Finally, OsRAC1 is involved in the OsCERK1/OsCEBiP-mediated immune signaling; it activates OsRbohB (OsNOX1) for ROS production by directly interacting with the N-terminus of NADPH oxidase [142,143]. Additionally, WAK2A-800 was shown to be positively involved in the responses to *Fg* through a chitin-induced pathway in wheat. More importantly, silencing TaWAK2A-800 reduced the expression levels of the chitin-triggering immune pathway marker genes TaCERK1, TaRLCK1B, and TaMAPK3 after inoculation with *Fg* [122]. For the mitogen-activated protein kinases (MAPKs), MAPK-mediated phosphorylation of WRKYs was a exciter, which promoted the downstream signalings: the activated WRKYs then bind to and regulate the expression of NOX/RBOH genes [107]. Just in time, one study reported that TaWRKY19 repressed plant immunity against pathogens by negatively regulating the transcriptional level of TaNOX10 and compromising ROS generation in wheat [144]. In addition, Köster et al. indicated that Ca^2+^ signaling was central to both pattern- and effector-triggered immunity activation of the immune system in plants [142]. Furthermore, the calcium-dependent protein kinase TaCDPK can directly interact with NOXs/RBOHs in a Ca^2+^-dependent manner [139,145,146], both of which are synergistically involved in plant defense responses to pathogens [147,148,149,150]. Therefore, under pathogen stimulation, plant cells often elevate the levels of ROS to implement plant immunity by eliminating compromised host cells, in turn limiting the further infection by the pathogen. At the same time, H_2_O_2_, the precursor of ROS, plays a crucial role during cell wall rigidification (lignification and crosslinking of cell wall monomers) [151]. On the other hand, plant cells also enhance the expression levels of peroxidase (APX), superoxide dismutase (SOD), and catalase (CAT) to scavenge ROS and maintain the redox balance in plant cells.

Based on these results, CERK1-mediated signaling models were constructed as shown in Figure 4, in which the regulation patterns of NOXs/RBOHs activity and maintenance of intracellular ROS homeostasis during plant response to FHB stress), will offer valuable information for further studies in this field and provide important cues for crop improvement by genetic engineering and molecular breeding during agricultural practices.

## 5. Biological and Chemical Control

### 5.1. Biological Control

In recent years, biological control techniques have been widely applied to control plant pathogens. There are many species of microorganism that can be applied in biological control, such as bacteria, fungi, and actinomycetes [20,152]. The main mechanisms of biocontrol microorganisms are antagonism, competition, and hyperparasitoidism. At present, a variety of antagonistic strains against *Fg* have been identified, such as *Bacillus subtilis*, *Pseudomonas radiobacter*, *Bacillus thuringiensis*, *Agrobacterium radiobacillus*, *Actinomycetes*, *Burkholderia yabunchirtal*, and the endophytic fungus *Simplicillium lamellicola* [19]. Among them, *Bacillus subtilis*, *Bacillus thuringiensis, Pseudomonas radiobacter*, and *Agrobacterium radiobacter* are easy to separate and cultivate; their dormant spores have strong stress resistance and long survival time. Accordingly, they have attracted people’s attention and have been commercialized as preparations, showing great application prospects in biological control [153]. For example, *Streptomyces* spp. from *Streptomyces Aureus* have a strong inhibitory effect on wheat FHB [154]. Zhang et al. (2020) found that *Streptomyces pratensis* strain S10 parasitized wheat roots and inhibited wheat FHB by inhibiting mycelial growth and reducing DON gene expression [155]. Moreover, Frenolicin B, the main active component from fermentation broth of *Streptomyces* sp. NEAU-H3 showed strong antifungal activity against *Fg* by affecting mycelia and cell contents [156]. In addition, *Bacillus velezensis* RC 218 was identified as a biocontrol fungicide against *Fg* by inducing cell wall thickening and preventing cell plasmolysis and collapse in the host [157]. Another research proposed that *Metarhizium anisopliae* was a potential biocontrol agent, of which 1% broth filtrate could impede conidial germination of *Fg*; *Metarhizium anisopliae* can combat *Fg* by producing secondary metabolites, which inhibit the fungi, promote wheat growth, and trigger a defense response in plants [158]. Xu et al. (2021) found that *Bacillus amyloliquefaciens* MQ01 reduced the pathogenicity of *Fg* by degrading zearalenone (ZEN) and synthesizing chitin-binding protein and *Bacillus subtilis* protease during antagonism [159]. Recently, a new report suggested that *Pantoea agglomerans* ZJU23, isolated from the bacterial microbiome of perithecia formed by *Fg*, could efficiently reduce fungal growth and infection by secreting a key antifungal metabolite, herbicolin A. Herbicolin A can destroy the lipid raft structure and cell membrane integrity by directly binding and disrupting ergosterol-containing lipid rafts, thereby inhibiting the growth, pathogenicity, and toxin synthesis of *Fg* [160]. In a word, these results illuminate that the mechanism of antagonistic strains against *Fg* operates mainly through the production of metabolites by inhibiting the growth of *Fg* mycelium, degrading pathogenic metabolites, and inhibiting toxin gene expression or promoting wheat growth to enhance FHB resistance.

In general, it is difficult for the mycelium to colonize and reproduce rapidly under harsh environmental conditions after germinating from spores. Moreover, the bioactivity of their metabolites is often limited by environmental conditions, which have become a key limiting factor for the development of plant growth-promoting microorganisms resistant to FHB in wheat. Therefore, screening and isolating beneficial microorganisms, unearthing multiple and pivotal FHB-resistance genes, and revealing their resistance mechanisms will help to breed powerful growth-promoting microorganisms by gene recombination technology. In summary, cultivating antagonistic microorganisms with broad-spectrum adaptability and strong resistance to FHB may become a trend in biocontrol studies associated with FHB.

### 5.2. Chemical Control

To date, there is very limited promising germplasm for wheat blast resistance in production, and chemical spraying during the flowering stage of wheat is the main method of FHB control. It has been demonstrated that the common chemical fungicides, including benzimidazole fungicide carbendazim, sterol demethylase inhibitor (DMI) fungicide tebuconazole, and mimosine are effective against FHB. Among these, carbendazim has a perfect inhibition effect and has been widely applied in agricultural production [161]. However, the single use of the same chemical fungicide over a long period of time inevitably leads to drug resistance of strains [162,163]. Therefore, with progress in research, a new group of antimicrobial fungicides such as cycloheximide, chlorothalonil, and prothioconazole has been studied and reported to cater to the needs of agricultural development. In addition, a chemical material, nanosilver, was confirmed as an inhibitor of the growth of pathogenic microorganisms such as bacteria, fungi, and mycoplasma; it is also highly lethal to certain viruses and protozoa without drug resistance and can be used as a long-lasting and safe fungistat [164]. Takemoto et al. (2018) found that K20, a novel amphiphilic aminoglycoside fungicide, had a good inhibitory effect on many fungal species, including *Fg* [165]. Duan et al. (2018) found that epoxiconazole inhibited the production of toxins such as DON by *Fg* [166]. Zhu et al. (2020) found that vitamin E had an indirect regulatory effect on the accumulation of vomitoxin (DON) in wheat [167]. Moreover, trans-2-hexenal, T2H, a typical green leaf volatile, synthesized from the primary metabolite linolenic acid or linoleic acid in plants, was proved to be a biofumigant for protecting crops against *Fg* [168]. Therefore, in the absence of resistant germplasm resources in wheat and with the scarcity of FHB-resistant strains with wide adaptability, the choice of chemical fungicides has become a common method of FHB control, but this is inevitably contrary to the concept of “advocating green production mode and promoting sustainable agricultural development”.

## 6. Conclusions and Perspectives

In summary, scientists worldwide have made great progress in FHB-resistance germplasm, genetics, and genomics in recent years, laying a solid foundation for the improvement of FHB-resistance breeding. However, the progress of resistance breeding is relatively slow, which cannot cater to the demand for resistance germplasm resources in agricultural production. The reasons for this are found mainly in the following aspects: (1) Although there are abundant FHB-resistance resources worldwide, they have not been studied and utilized widely, due to their narrow genetic base and poor agronomic traits. (2) Although some QTLs/genes for resistance to FHB have been discovered, the number of resistance master genes that have been cloned is limited, and their function and stability have rarely been studied and validated in depth. (3) Research on FHB-resistance mechanisms is mainly at the morphological and physiological levels, and little research is at the molecular level. In addition, important agronomic traits such as wheat yield and quality are complex quantitative traits influenced by multiple genes and environmental interactions. Accordingly, it is difficult to cater to food security and needs for a better life by relying solely on conventional breeding techniques. Therefore, it is imperative to accelerate research on wheat genomic and molecular genetic breeding. Screening for more and better resistance master genes, mapping their fine genetic profiles, and evaluating their practical application are the urgent tasks for us now.

## Figures and Tables

**Figure 1 cells-11-02275-f001:**
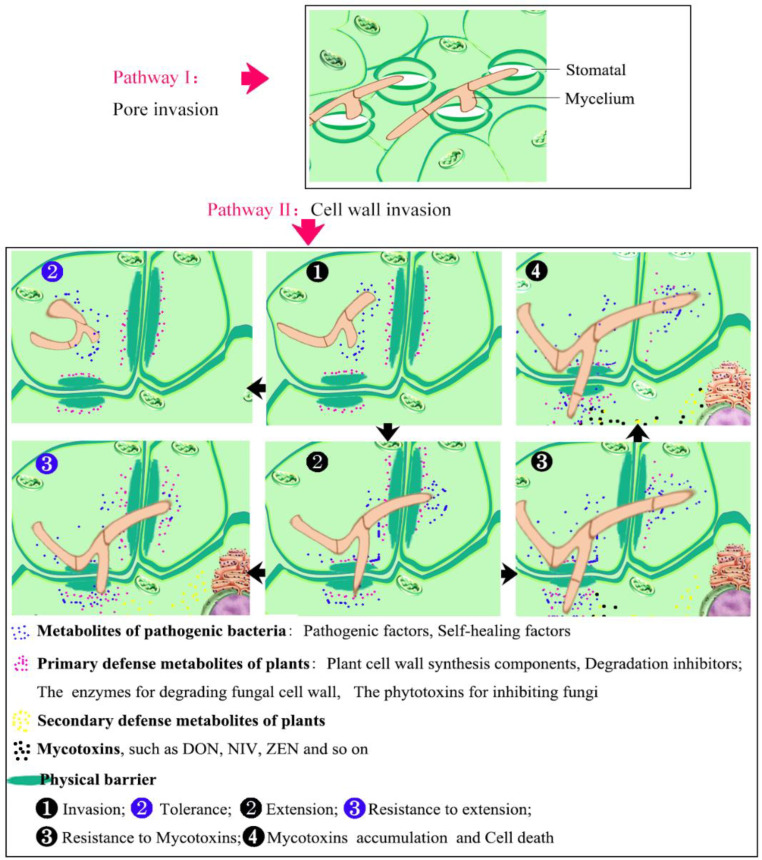
Model of the war between host and FHB-related pathogen. There are two ways of infecting plants with FHB-related pathogen. Pathway I: Pore invasion, in which the mycelium invades the plant directly through the natural pores present in the plant, such as stomata, the opening of the apical floret, and the gap between the palea and the outer glume. Pathway II: Cell wall invasion, where the pathogenic mycelium invades the plant directly or expands within the cells by destroying and degrading the cell surface cuticle and the cell wall. ❶ The fungus hydrolyzes the plant cuticle by synthesizing some hydrolytic enzymes (e.g., keratinase) and repairs its own cell membrane and cell wall by synthesizing chitin synthase, sphingolipids, and glucosylceramide synthase to counteract the defensive damage from the plant, while the plant reinforces the physical barrier formed by the cuticle and cell wall by synthesizing lignin, wax lipids, and pectin. Meanwhile, the plant also synthesizes chitinases, tylopectinase, and glucanase to degrade the fungal cell wall and synthesizes thionins, non-specific lipid transfer proteins, puroindolines, and ROS to disrupt and damage the fungal membrane integrity, finally counteracting the fungal invasion. ❷ The plant has stronger resistance, and the physical barrier can prevent the expansion of the pathogenic mycelium and shows a state of tolerance to the invading pathogen. ❷ The pathogenic mycelium breaks through the physical barrier of the plant and infests the adjacent cells. At this point, the plant will secrete some phytotoxins (e.g., isohydroxamate phenols) and/or synthesize phenols and bacteriostatic proteins to further inhibit the growth of the mycelium and synthesize polygalacturonase and xylanase to inhibit the degradation of its own cell wall. The pathogen further synthesizes pectinases, hemicellulases, cellulases, lipases, and nucleases to damage plant cell membranes and cell walls. ❸ The invasion of the pathogen promotes the expression of some resistance proteins from the plant to inhibit the development of the mycelium and the production of toxins, and the plant behaves as a tolerant. ❸ Under pathogen stimulation, plant cells elevate the level of ROS and polyamines, which promote oxidative damage in fungal cells. On the other hand, when ROS accumulation in plant cells reaches a certain level, this will trigger the expression of oxidases, such as peroxidase (APX), superoxide dismutase (SOD), and catalase (CAT), to scavenge ROS and reduce oxidative damage for themselves and inhibit intracellular toxin accumulation. ❹ At later stages, the pathogen spreads wildly and triggers a strong ROS production, which, in turn, promotes disease spread and toxin accumulation and then results in programmed cell death in the plant.

**Figure 2 cells-11-02275-f002:**
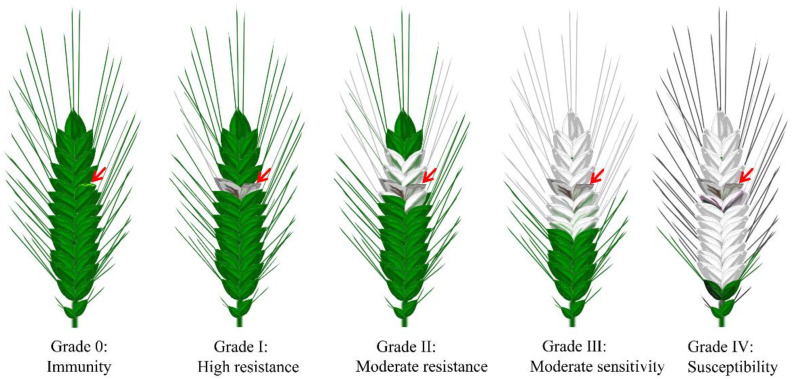
Grades of resistance to and severity of FHB in wheat. According to the severity of FHB on wheat spikes, the severity of the disease and the resistance level of the plant were divided into five grades: grade 0, I~IV, which represents the percentage of diseased spikelets as zero, less than 1/4 (<25%), 1/4~1/2 (26–50%), 1/2~3/4 (51–75%), and more than 3/4 (>75%) in turn. Meanwhile, the numbers also correspond to the different FHB resistance levels of the plant. (Please refer to the detailed information in GB/T 1576-2011 of China). Red arrows indicate sites inoculated with *Fg*.

**Figure 3 cells-11-02275-f003:**
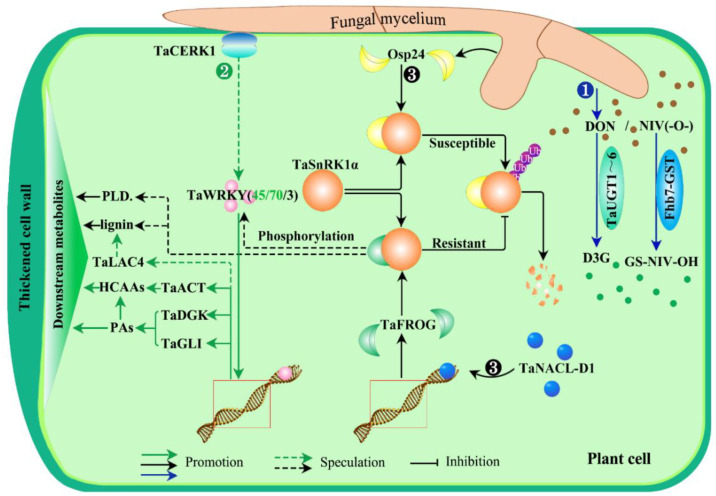
FHB-resistant signaling pathways in fungus–plant interactions. There are three resistant pathways: first, the detoxification pathway, in which the members of (UDP)-glucose transferase TaUGT1~6 can enhance plant tolerance to FHB by converting the fungal mycotoxin DON to the less toxic DON-3-O-glucoside, D3G (❶). In addition, another gene, Fhb7, encoding a glutathione S-transferase (GST), from a distant variety of wheat, *Elytrigia elongata*, confers broad resistance and tolerance to *Fusarium* species by detoxifying trichothecenes, such as NIV, through conjugating GSH to the epoxy group of NIV. Second are CERK1-mediated metabolic pathways to thicken the plant walls. It is well known that the first layer of innate immunity in plants is initiated by the surface-localized pattern recognition receptors (PRRs), in which the leading role is played by chitin elicitor receptor kinase 1 (CERK1), which mainly participates in chitin-induced immunity [107]. Meanwhile, chitin is a typical component of the fungal cell wall and can trigger the plant innate immune response by activating PAMPs (pathogen-associated molecular patterns). Here, a transcription factor, HvWRKY23, can be induced by HvCERK1 with some unclear methods upon infection with *Fg*. HvWRKY23 subsequently regulates downstream genes, such as *HvPAL2*, *HvCHS1*, *HvHCT*, *HvLAC15*, and *HvUDPGT*, to biosynthesize HCAAs, flavonoid glycosides, lignin, and so on, which reinforces the cell walls to contain the spread of *Fg* in plant cells. Similarly, TaWRKY70 was also identified as having a potential role against *Fg* in the early stages of defense through physically interacing with and activating the downstream genes *TaACT*, *TaDGK*, and *TaGLI*, among which TaACT is the rate-limiting enzyme in the biosynthesis of HCAAs and TaDGK and TaGLI are the crucial enzymes in the biosynthesis of PAs (phosphatidic acids) in plants [108]. Moreover, HCAAs and PAs have functions during fungus–plant interactions; PAs contribute to the plant defense response through translocation and catabolism in the apoplast, leading to the production of H_2_O_2_, which has several roles, such as directly eliminating pathogens and assisting in cell wall strengthening; HCAAs were identified as biomarkers during fungus–plant interactions and support a functional role in plant defense [109]. What is more, a guanidine cinnamyl transferase gene, TaACT, was confirmed as having a role involved in FHB response in plants [110]; another potential downstream gene, *TaLAC4*, encoding a wheat laccase, was also reported to restrain FHB infestation by promoting the synthesis of lignin in the secondary cell wall of plants in order to thicken the cell wall [100]. Therefore, this indicated that a signaling pathway of WRKY-mediated RRI metabolite accumulation in cells perhaps plays an important role in the plant response to FHB stress (❷). The second is an energy-related metabolic pathway. It is known that many pathogens have acquired the ability to inject virulence effector proteins into host cells to achieve more effective infection. As shown in ❸, FHB-causing fungi secrete a virulence effector protein, Osp24 (an orphan protein), which promotes mycelium growth and toxin accumulation in plant cells. At the same time, Osp24 can combine with TaSnRK1α, a FHB-resistant protein, resulting in TaSnRK1α being ubiquitinated and degraded. There is always an arms race between the pathogen and its host. For plants, the second layer of immune recognition is intracellular immune receptor forms, and the intracellular immune receptors are most often the nucleotide-binding proteins, which can recognize those effectors and elicit a second layer of defense. From this, it is not difficult to speculate that the N-acetylcysteine-like transcription factors, TaNACs (TaNACL-D1/TaNAC032), as the intracellular immune receptors, perhaps can recognize Osp24 or its analogs, and then are activated by them. Subsequently, the activated TaNACs regulate the expresion of the downstream gene TaFROG, which encodes a wheat orphan protein (an analog of Osp24). Thus, TaFROG can compete against Osp24 to combine with TaSnRK1α, preventing TaSnRK1α from being ubiquitinated and increasing plant resistance to FHB. However, the resistance mechanism of TaSnRK1α to FHB in plants still remains unclear. In addition, Han et al. found that SnRK1 interacts with and phosphorylates WRKY3 repressor in barley, leading to the degradation of WRKY3 and enhanced barley immunity [111]. Consequently, there may be molecular crosstalk between pathway ❷ and ❸ mediated by phosphorylation. PLD: phenylpropanoid; HCAAs: hydroxycinnamic acid amides; PAs: phosphatidic acids; TaACT: agmatinecoumaroyl transferase; TaDGK: diacylglycerol kinase; TaGLI: glycerol kinase.

**Figure 4 cells-11-02275-f004:**
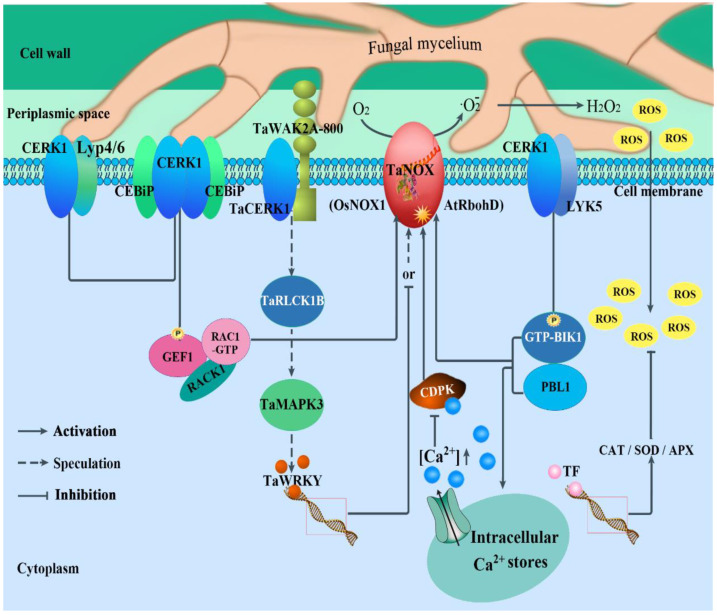
ROS homeostasis regulated by CERK-mediated immune signaling plays a crucial role during fungus–plant interactions. In this figure, the chitin-induced particles CERK, LysM, CEBiP, BIK1, and ROPs/RACs and GEF-mediated ROS homeostasis by regulation of NOX activity, perhaps perform vital roles in plant response to FHB. In Arabidopsis, a chitin elicitor receptor kinase CERK1, and a LysM-containing receptor-like kinase LYK5 were found to be able to bind chitin and form a chitin-dependent immune complex, CERK1-LYK5 [132,133], in which AtCERK1 is involved in activating BIK1 by phosphorylation directly. The activated BIK1 (GTP-BIK1) directly phosphorylates AtRbohD and enhances ROS production for defense responses [134,135]. In rice, a plasma membrane-localized receptor-like kinase OsCERK1 is associated with receptor-like kinase OsCEBiP under chitin treatment [136]. In addition, OsCEBiP can form a homodimer upon chitin binding that is followed by heterodimerization with OsCERK1, creating a signaling-active sandwich-type receptor system [137,138]. Then, OsGEF1, a regulator functioning as a molecular switch, is phosphorylated by OsCERK1 [139]. The activated OsGEF1, coupling with OsRACK1 [140] in turn, phosphorylates a ROP/RAC GTPase, OsRAC1, and changes it from the inactive form RAC-GDP to the active form RAC-GTP [130,141]. Finally, the actived OsRAC1 then activates OsRbohB (OsNOX1) for ROS production by directly interacting with the N-terminus of NADPH oxidase [142,143]. Additionally, a wall-associated kinases WAK2A-800 performed a positive role in response to *Fg* through a chitin-induced pathway. More importantly, silencing TaWAK2A-800 reduced the expression levels of the chitin-triggering marker genes TaCERK1, TaRLCK1B, and TaMAPK3 under treatment with *Fg* [121]. MAPK-mediated phosphorylation of WRKYs was a exciter, which promoted the downstream signalings: the activated WRKYs then bind to and regulate the expression of NOX/RBOH genes [107].For example, TaWRKY19 repressed plant immunity against pathogens by negatively regulating the transcriptional level of TaNOX10 and compromising ROS generation in wheat [144]. In addition, Ca^2+^ signaling was central to both pattern- and effector-triggered immunity activation of the immune system in plants [142]. Moreover, CDPKs, in a Ca^2+^-dependent manner, can directly interact with and activate NOXs/RBOHs for ROS production, which plays a crucial role in plant defense responses to pathogens. Therefore, under pathogen stimulation, plant cells often elevate the levels of ROS to implement plant immunity by eliminating compromised host cells, in turn limiting the further infection by the pathogen. At the same time, H_2_O_2_, the precursor of ROS, plays a crucial role during cell wall rigidification (lignification and crosslinking of cell wall monomers) [151]. On the other hand, plant cells also enhance the expression levels of peroxidase (APX), superoxide dismutase (SOD), and catalase (CAT) to scavenge ROS and maintain the redox balance in plant cells. CERK1: a plasma membrane-localized chitin elicitor receptor kinase 1; LYP4/6: receptor-like OsLYP4 and OsLYP6; CEBiP: receptor-like kinases; LKY5: LysM-containing receptor-like kinase 5; GEF: guanine nucleotide exchange factors; RAC1-GTP: a subfamily of Rho-type GTPases; RACK: receptor for activated C-kinase; CDPK: calcium-dependent protein kinase; BIK1: botrytis-induced kinase 1; PBL1: PBS-like kinase, which is a close homolog of receptor-like cytoplasmic kinases; TF: transcription factor.

**Table 1 cells-11-02275-t001:** Information on the resistance QTLs to FHB in wheat.

QTL Name	Flanking Marker	on Chr. *	Resistance Type	Representative	Reference
Fhb1Qfhs.ndsu-3AS	Xwgc501–Xwgc510	Chr. 3AS	Type II	*Triticum dicoccoides*	[88]
Fhb1(Qfhs.njau-3B)	Xmag8937–Xmag9404	Chr. 3BS	Type II	Sumai 3	[89]
Fhb1(Qfhs.ndsu-3BS)	XGWM533–XGWM493	Chr. 3BS	Type II	Sumai 3	[90]
Fhb1(TaHRC)	TaHRC-GSM/TaHRC-Kasp	Chr. 3BS	Type II	Sumai 3	[91]
Fhb2	Xwmc398–Xgwm644	Chr. 6BS	Type II	Sumai 3	[92]
Fhb3	BE586744-STS–BE586111-STS	Chr. 7ALr#1S	Type II	Sumai 3	[50]
Fhb4(Qfhi.nau-4B)	Xhbg226–Xgwm149	Chr. 4B	Type I	Wangshuibai	[93]
Fhb5(Qfhi.nau-5A)	Xbarc56–Xbarc100	Chr. 5A	Type I	Mianyang 99-323	[94]
Fhb6	BF202643/HaeIII-BE591682/HaeIII	Chr. 1E	Type II	Sumai 3	[51]
Fhb7	XsdauK86–XsdauK88	Chr. 7E/7D	Type V	*Elytrigia elongata*	[95]
Qfhs.nau-2A	Xwmc474–Xsm1021	Chr. 2A	Type II	Nanda 2419	[96]
Qfhi.nau-2B	Xwmc499–Xmag1729	Chr. 2B	Type I	Nanda 2419	[26]
Qfhs.nau-2B	Xmag1811.1–Xmag3080	Chr. 2B	Type II	WSB	[96]
QFhb.nau-2B	Xwgrb1561–Xwgrb1420	Chr. 2B	Type I/Type II	Nanda 2419	[90]
Qfhi.nau-3A	Xwmc169–Xgwm162	Chr. 3A	Type I	WSB	[26]
QFhb-hnau.3BS.1	gwm533–stm748tcac	Chr. 3BS	Type I/Type II	Landrace N553	[25]
Qfhs.nau-3B	Xgwm389–Xbarc102	Chr. 3B	Type II	WSB	[96]
Qfhi.nau-4A	Xwmc161–Xmag3886	Chr. 4A	Type I	Nanda 2419	[26]
Qfhi.nau-4B	Xgwm495–Xgwm149	Chr. 4B	Type I	WSB	[97]
Qfhs.ifa-5A	barc186–wmc805	Chr. 5A	Type I/Type II	Chinese spring	[98]
QFhb-5A	Xgwm304–Xgwm415	Chr. 5AS	Type II	Yangmai 158	[55]
Qfhi.nau-5A	Xbarc180–Xgwm186	Chr. 5A	Type I	WSB	[97]
Qfhs.nau-6B	Xwmc398–Xmag359	Chr. 6B	Type II	WSB	[96]
Qfhi.nau-2D	Xwmc181–Xaf12	Chr. 2D	Type I	WSB	[26]
QFhb-hnau.2DL	AX-110955068–AX-109419238	Chr. 2DL	Type I/Type II	Yangmai 13	[25]
QFhb.cau-7DL	gwm428	Chr. 7DL	Type II/Type I	Sumai 3	[99]

* Chr. represents chromosome; Italics represents the Latin format for species.

**Table 2 cells-11-02275-t002:** Information on the resistance genes to FHB in wheat.

Nomenclature	Protein	Gene ID	Location on Chr. */ Flanking Markers	Function
FHB7-GST	GST: Glutathione transferase	Tel7E01T1020600.1	Chr. 7E/7D (XsdauK86, XsdauK88)	A FHB-resistance gene, *Fhb7*, from *Thinopyrum elongatum*, introgressions artificially in wheat, confers resistance to FHB without yield penalty [95].
LTP	Lipid transfer protein	Ta.1282.4.S1_at	Chr. 5A In the interval of Qfhs.ifa-5A.	LTPs might confer plant type I resistance against initial fungal penetration of *Fg* and also contribute to toxin resistance [98].
MIN7	ARF-GEF protein	TraesCS2A02G202900 TraesCS2B02G230000 TraesCS2D02G212800	Chr. 2	TaMIN7 plays positive role, involved in response to *Fg* by disturbing vesicular trafficking in cells [112].
Qfhs.ifa-5A (Fhb1)	Lipid transfer protein	GenBank: FN564434	Chr. 3BS	Confers type II resistance [98].
Qfhs.njau-3B	Histidine-rich calcium-binding protein	GenBank: KX022627.1–KX022633.1, MK397611–MK397761 (His: Xmag8937)	Chr. 3BS Xmag8937 and Xmag9404	Qfhs.njau-3B is a candidate gene on Fhb1, encodes a histidine-rich calcium-binding protein, mutation of it in wheat confers resistance to FHB [89].
TaAOS	Oxidized propadiene synthase	TraesCS4A02G061900 TraesCS4D02G238800 TraesCS4B02G237600	Chr. 4A/4D/4B	TaAOS involved in JA signaling pathway to enhance plant resistance to FHB. TaAOS-silenced strains exhibit high susceptibility to FHB [70].
TaACT	Agmatine coumaroyl transferase	GenBank: KT962210	Chr. 2DL (FHB *QTL-2DL*) (WMC245-GWM608)	TaACT is an important gene in wheat FHB QTL-2DL, conferring type II resistance to *Fg* by limiting the spread of pathogen from the initial point of infection [110].
TaFROG	Orphan protein	GenBank: KR611570	Chr. 4A (CM82036)	TaFROG binds to the protein TaSnRK1α to prevent TaSnRK1α from being degraded by ubiquitination, thereby increasing the resistance of the plant to FHB [113].
TaHRC (TaHRC-S) (TaHRC-R) (Fhb1)	Histidine-rich calcium-binding protein	GenBank: CBH32655.1 GenBank: MK450309 GenBank: MK450312	Chr. 3BS in the interval of QTL-Fhb1 (syn Qfhs.ndsu-3BS; Gwm533,Gwm493)	TaHRC, a key factor of fhb1-mediated FHB resistance, encodes a nuclear protein (histidine-rich calcium-binding protein) with FHB susceptibility. Mutating *TaHRC* in plants leads to increased resistance to FHB [52].
TaJRL53	Jacalin-related lectins protein	TraesCS4A02G430200.1	Chr. 4AL	TaJRL53 enhanced FHB resistance in wheat through regulating ROS synthesis pathway and JA signal transduction pathways [114].
TaLAC4 (Fhb1)	Laccase	TraesCS3B02G392700.1	At QTL-Fhb1 on Chr. 3BS (syn Qfhs.ndsu-3BS; WMC245,GWM608)	TaLAC4, a wheat laccase, can increase resistance of plants to FHB by promoting the synthesis of lignin in the secondary cell wall to thicken the cell wall and confers type II resistance [100].
TaNACL-D1	N-Acetylcysteine-like transcription factors	TraesCS5D02G111300 (GenBank: MG701911)	Ch. 5D	TaNACL-D1 interacts with the orphan protein TaFROG to improve the resistance of the plant to FHB [107].
TaNAC032 (Fhb1)	Lignin-biosynthetic	GenBank: MT512636	At QTL-Fhb1 (syn Qfhs.ndsu-3BS) *XSTS3B*-80 and *XSTS3B*-142	Promotes transcription of lignin genes associated with resistance-related metabolite biosynthesis [101].
TaPFT	Pore-forming toxin-like	GenBank: KX907434.	Chr. 3BS At QTL-Fhb1 (syn Qfhs.ndsu-3BS)	PFT, encoding a perforatoxin analog, is from Fhb1 on the genome of Sumai 3 and shows significant resistance to FHB [68,69].
TaRBL	Ricin B-like lectin protein	TraesCS3A02G078200.1 TraesCS3B02G093100.2 TraesCS3D02G078700.1	Chr. 3	TaRBL interacts with TaPFT and is involved in resistance to FHB in wheat [115].
TaSHMT3	*Serine hydroxy methyltransferase*	TraesCS3A02G385600	Chr. 3A	Silencing of TaSHMT3A-1 compromises Fusarium head blight resistance in wheat [116].
TaSnRK1α	The alpha subunit of sucrose non- fermentation-related kinase 1 (SnRK1)	GenBank: KR611568	Chr. 4A (CM82036)	TaSnRK1α interacts with TaFROG to prevent degradation by ubiquitination and improves the resistance of the plant to FHB [113,117].
TaSSI2-2AL -2BL -2DL	*Stearoyl-acyl carrier protein fatty acid desaturase*	AA0283540, AA0388780 and AK332689	Chr. 2	TaSSI2 probably regulated FHB resistance by depressing the SA signaling pathway in wheat [118].
TaUGT3 (Fhb1)	UDP-glycosyltransferase	GenBank: FJ236328	Chr. 3BS	Ta-UGT 3 was found to enhance host tolerance against deoxynivalenol (DON) in *Arabidopsis* [104].
TaUGT5	UDP-glycosyltransferase	Ta_iwgsc_2bs_vl_5195782	Chr. 2B	TaUGT5 can reduce the proliferation and destruction of *Fg* and enhance the ability of FHB resistance in wheat [102].
TaUGT6	Glycosyltransferases	TraesCS5B02G436300	Chr. 2B	The protein TaUGT6 can transform the toxin DON (deoxynivalenol) into non-toxic D3G (D3-glucoside); O\overexpression of TaUGT6 in wheat enhanced the resistance of plant to *Fg* [106].
TaUGT12887	Glycosyltransferases	TraesCS5B02G148300 GenBank accession JX624788	In the interval of *QTL* Fhb1	Confers plant weak resistance against DON [98].
TaWRKY45	WRKY-like transcription factor	EMBL/GenBank accession numbers AB603888, AB603889; AB603890	Chr.2	Overexpression of the TaWRKY45 transgene conferred an enhanced resistance against *Fg* in wheat [119,120].
TaWRKY70	WRKY-like transcription factor	TraesCS2D02G489700	In the interval of *QTL-2DL*	TaWRKY70 silencing lines not only increased fungal biomass but also decreased expressions of downstream resistance genes TaACT, TaDGK, and TaGLI1, along with decreased abundances of RRI metabolites biosynthesized by them [108].
TaWAK2A-800	Wall-associated kinase	TraesCS2A02G071800.1	Chr.2A	TaWAK2A-800 participates positively in the resistance responses to *Fg*, possibly through regulating the chitin-triggering immune pathway marker genes, TaCERK1, TaRLCK1B, and TaMPK3 in wheat [121].
UDP-UGT	Uridine diphosphate (UDP)- glucosyltransferase (UGT)	Traes_2BS_14CA35D5D	Chr. 2B	Improving the resistance of plant to FHB and the tolerance to DON as well as potentially conjugating DON into D3G in plants [105].
WFhb1-1	Putative membrane protein	GenBank # KU304333.1)	In the interval of Qfhb1	Overexpressing *WFhb1-1* in non-Qfhb1-carrier wheat led to a significant resistance to *Fg* [122].

* Chr. represents chromosome, *Fg* represents the abbreviation of *F. graminearum*; Italics represents the Latin names of species.

## Data Availability

Data sharing not applicable. No new data were created or analyzed in this work.

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
