# Peer review of "Arms Race between the Host and Pathogen Associated with Fusarium Head Blight of Wheat"

_cells, 2022, doi:10.3390/cells11152275_

Round 1

Reviewer 1 Report

1.   The title is suspicious. Based on published data, arms race seldom ocurr between fusarium species and its host just since the resistance gene for fusarium head blight may not be a typical R gene.

2.   I understand the authors input lots of efforts into this review paper,unfortunately,this paper seems to stack everything instead of providing critical and knowledgeable information on specific topic to potential readers and researchers.

3.   Too many errors or inaccurate descriptions throughtout the text.

Author Response

Thank you very much for your work and your opinions on our manuscript. Please see the attachment for the detailed information.

Reviewer 2 Report

Congratulations to the authors of the paper.  GOOD WORK!.  I am returning the original with some corrections that deserve to be considered for improving its quality.

Author Response

(The authors gave the same response as above.)

Reviewer 3 Report

The paper “Arms Race Between the Host and Pathogen Associated With Fusarium Head Blight”, by Hu and co-authors, is a review on the FHB, probably the most destructive wheat disease. The paper deals with the recent advances on pathogenesis, resistance-associated QTLs/genes, resistance mechanisms, signaling pathways and control methods.

The paper is quite clear and well written but there are several grammatical errors throughout the text that often produce misunderstanding of the sentences, this aspect should be carefully checked before publication, together with the following suggestions:

Line 51-69: add some references dealing with the listed points

Line 72-74: the cited article is of 2006, not really a “recent year” as stated. Add more recent references

Line 294: replace analysis with studies

Lines 482-484: revisions are needed for several species names

Lines 511-522: the meaning is not clear, please revise the sentences and add some references

Line 530: antibacterial?

Line 538: not clear

Line 545: bionic?

Lines 575-617 and fig. 4: this should not be included in Conclusions and Perspectives, it should be a separate chapter on ROS homeostasis or something similar, and it should integrates also what reported in the legend of fig.4

Author Response

(The authors gave the same response as above.)
